# Importance of Apolipoprotein A-I and A-II Composition in HDL and Its Potential for Studying COVID-19 and SARS-CoV-2

**DOI:** 10.3390/medicines8070038

**Published:** 2021-07-16

**Authors:** Kyung-Hyun Cho

**Affiliations:** 1Medical Innovation Complex, Korea Research Institute of Lipoproteins, Daegu 41061, Korea; chok@yu.ac.kr; Tel.: +82-53-964-1990; Fax: +82-53-965-1992; 2LipoLab, Yeungnam University, Gyeongsan 712-749, Korea

**Keywords:** COVID-19, SARS-CoV-2, high-density lipoproteins, apoA-I, apoA-II, glycation

## Abstract

The composition and properties of apolipoprotein (apo) A-I and apoA-II in high-density lipoproteins (HDL) might be critical to SARS-CoV-2 infection via SR-BI and antiviral activity against COVID-19. HDL containing native apoA-I showed potent antiviral activity, while HDL containing glycated apoA-I or other apolipoproteins did not. However, there has been no report to elucidate the putative role of apoA-II in the antiviral activity of HDL.

Coronavirus disease 2019 (COVID-19) is an ongoing pandemic caused by the severe acute respiratory syndrome coronavirus 2 (SARS-CoV-2). In March 2020, the World Health Organization (WHO) declared the outbreak a pandemic. In November 2020, Wei et al. published an interesting article reporting that scavenger receptor B-I (SR-B1) facilitates the ACE-2-dependent entry of SARS-CoV-2 [1]. SARS-2-S is a spike (S) protein of SARS-CoV-2, around 180–200 kDa in size, that can be cleaved into the S1 (SARS-2-S1) and S2 (SARS-2-S2) subunits, which are responsible for receptor recognition and membrane fusion, respectively. They also found that the S1 subunit of SARS-2-S could bind to cholesterol and possibly other HDL components to enhance viral uptake in vitro. On the other hand, however, they found that SARS-2-S did not bind to apoA-I, the major protein constituent of HDL. The authors reported that they used commercially available HDL (Sigma L8039) in their experiments [1]. With several surface plasmon resonance data, they concluded that the SARS-2-S1 has specific affinity for cholesterol and possibly HDL components. Although the components were not identified explicitly, they reported that HDL enhances the entry of SARS-CoV-2 through the SARS-2-S1 protein.

Although the in vitro study cannot be directly compared with other clinical cases, these results are questioned by many clinical reports regarding the human lipid profile after the onset of COVID-19. Patients with COVID-19 in Wenzhou, China, showed a remarkable decrease in HDL-C and an elevation of the monocyte/HDL-C ratio, particularly in subjects with a primary infection and male patients [2]. Soon after an infection, most patients showed a rapid decrease in serum cholesterol level, which was positively correlated with COVID-19 severity [3]. In particular, the HDL-C levels decreased according to the severity of the cytokine storm [4]. HDL-C is an important factor affecting virus clearance in COVID-19 patients [4]. In patients with SARS-CoV-2 nucleic acid long-term positivity (>14 days), low serum HDL-C levels were significantly associated with longer clearance, approximately 30 days after the onset of COVID-19 [5]. Other studies have shown that the average time from the beginning of symptom onset to the first negative test of a throat swab SARS-CoV-2 was 9.5–11 days [6]. These results strongly suggest that high HDL-C levels might be beneficial in symptomatic patients with COVID-19 via putative antiviral activity. An epidemiological study, the UK Biobank cohort study [7], reported that high serum HDL-C levels were associated with protection against hospitalization for COVID-19 and its mortality.

There are remarkable discrepancies between the in vitro viral entry hypothesis and clinical changes in serum HDL-C. The abrupt decrease in HDL-C upon the onset of COVID-19 [2] is difficult to explain if HDL-C enhances the viral entry after a SARS-CoV-2 infection [1]. Furthermore, the reasons why the elderly, male smokers, and patients with underlying diseases are infected more easily by SARS-CoV-2, despite having low HDL-C levels, are difficult to explain. Additionally, females, who usually show higher serum HDL-C levels than males, present a lower infection rate of COVID-19. If normal and native HDL mediates viral entry, young and healthy people should be infected more easily by SARS-CoV-2 than old subjects and patients with comorbidities affecting HDL composition and function. These clinical facts are in good agreement with the native HDL exerting a broad spectrum of antiviral activity against various viruses, as described for vaccinia, poliovirus, and herpes simplex virus [8]. On the other hand, the antiviral activity of HDL is contradicted by the putative interactions between HDL and SARS-CoV-2 to facilitate viral entry [1]. However, there are several differences between the two studies, with a major difference being the presence of apoA-II and glycation extent in commercial HDL. The Wei group used commercial HDL (Sigma L8039) and Huh7 cells (0403, hepatoma cell line from the Japanese Collection of Research Bioresources). 

Our recent study demonstrated that native HDL without apoA-II (lane 1, Figure 1A) displayed antiviral activity against SARS-CoV-2 (EC_50_ = 52.1 ± 1.1 μg/mL, final 1.8 μM), while glycated HDL (lane 2, Figure 1A) lost its antiviral activity [9]. The structure and composition of native HDL reflects its functionality, such as its antioxidant activity, which is very important for maximizing antiviral activity. Large HDL particles with high paraoxonase-1 activity showed stronger antiviral activity and cell viability. The greater glycation extent of HDL was associated with more atherogenic properties and cytotoxicity of Vero E6 cells [9]. These results may explain why patients with diabetes mellitus or hypertension are more susceptible to SARS-CoV-2 infection and have a higher risk of COVID-19 mortality. Similarly, a recent proteomic study also confirmed that PON-1 expression in HDL was markedly lowered in COVID-19 positive patients in an intensive care unit [10]. Accordingly, HDL isolated from severe COVID-19 patients displayed a loss of endothelial, anti-inflammatory protective effects, underlying the importance of HDL composition in maintaining HDL functionality.

In order to address and evaluate the HDL quality used by Wei et al. in their study, I asked for more information from Sigma-Aldrich regarding the lot used in the study [1] (Cat # L8039). With the lot number (SLCF0413), which was provided by the authors, certificates of analysis and origin were obtained from Sigma-Aldrich with SDS-PAGE data (Figure 1B). Unexpectedly, there were two bands around 25 kDa (major) and 10 kDa (minor). According to a technical service report by Sigma-Aldrich, the major band is most likely apoA-I, as described by the manufacturer, even though the molecular weight was smaller than native apoA-I (28 kDa). The minor band is apoA-II, as claimed by the manufacturer according to the electrophoretic profile (12% Tris-Glycine gel). ApoA-II is homo-dimeric protein around 17.4 kDa and it can be reduced to monomeric protein (around 8.7 kDa). Densitometric image analysis showed that the minor band comprised 25–30% of the total band area and intensity. ApoA-I and apoA-II comprise around 70% and 20% of total HDL protein content, respectively. ApoA-II is the second most abundant protein constituent of HDL.

At any rate, depending on the batch, the commercial HDL contained a large protein band of apoA-II other than apoA-I (Figure 1B), while native HDL from healthy Korean subjects showed almost no minor band, as shown in a recent report [9].

As the minor band is apoA-II, it might modulate the interaction between HDL and SR-BI. The elevation of apoA-II content in HDL is inversely associated with HDL binding and selective CE uptake by SR-BI [11]. Although serum apoA-II level can be changed with age, health state, and ethnicity, strenuous exercise could lower apoA-II and raise apoA-I and paraoxonase level [12].

The association between apoA-II and the risk of coronary artery disease (CAD) remains controversial, while it has been firmly established that the apoA-I levels are inversely associated with the risk of CAD. Human apoA-II was also shown to displace apoA-I in a reconstituted HDL, which impaired HDL particle functionality [13]. Moreover, apoA-II enrichment in HDL displaces paraoxonase from HDL and disrupts or inhibits its antioxidant properties [14]. Although apoA-II is an enigmatic protein, it is considered an atherogenic factor that hinders the beneficial functions of HDL, including its antioxidant ability. De Beer et al. suggested that the presence of apoA-II modulates the binding and selective lipid uptake of HDL by SR-BI [15]. These results might influence the different facilitation of viral entry via SR-BI because apoA-II has shown different kinetic stabilization of and binding affinity for HDL [16]. Moreover, apoA-II showed pro-inflammatory activity via suppressing the inhibitory activity of lipopolysaccharide-binding protein [17]. Taken together, these reports suggested that a decline in apoA-II level in serum and HDL may help to enhance the anti-infection, anti-inflammatory, and anti-viral activity of HDL [18]. 

**Figure 1 medicines-08-00038-f001:**
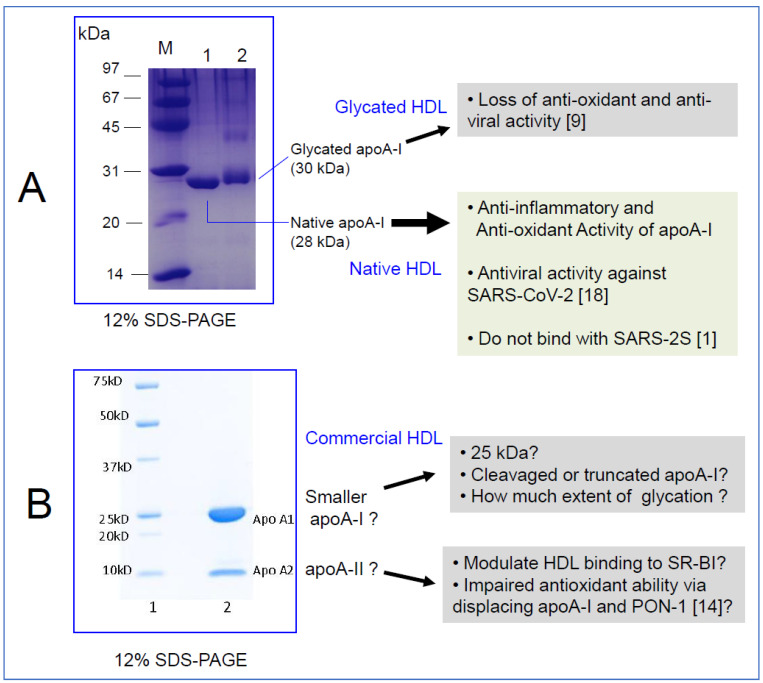
Different properties of apoA-I and apoA-II in HDL from different sources. Image of SDS-PAGE stained by Coomassie blue. (**A**) Electrophoretic patterns of native HDL (lane 1), purified from healthy and young Korean subjects by sequential ultracentrifugation. Glycated HDL (lane 2) was fructosylated for 72 h. M, molecular weight standard (Bio-Rad, low range, Cat # 1610304). (**B**) Electrophoretic patterns of commercial HDL (Sigma L8039, lot # SLCF0413). This gel photograph (12% Tris-Glycine gel, Invitrogen cat # XP00120) was courteously provided by Research & Applied Solutions of Merck/Sigma-Aldrich with permission. Lane M, molecular weight standard. Precision Plus Protein Unstained Standards (Bio-Rad cat # 161-0363).

Elderly groups (people in their 70s) have shown smaller HDL particle sizes relative to younger subjects (people in their 20s) [19]. Furthermore, young and healthy subjects showed 28 kDa of apoA-I in HDL, while elderly subjects showed 25–26 kDa of apoA-I due to the fragmentation of apoA-I [20]. The senescence-related truncation occurred at the N-terminal and C-terminal of apoA-I with an increase in the glycation extent in HDL. Overall, the smaller molecular weight of apoA-I in HDL may be associated with the production of dysfunctional HDL [19,21].

Elderly subjects and patients with underlying diseases, such as hypertension, diabetes, or coronary heart disease, usually show a higher risk of COVID-19 and lower levels of apoA-I with truncation and multimerization. The impairment of apoA-I might be associated with a higher risk of COVID-19 and the slower removal and clearance of SARS-CoV-2, as shown in Figure 1A. Compared to commercial HDL, native HDL (lane 1, Figure 1A), which exerts antiviral activity against SARS-CoV-2, did not show a minor band, as recently reported [9].

There might be a remarkable difference in the serum levels of apoA-II between ethnic groups depending on age and ongoing disease. HDL containing apoA-I alone (LpA-I) and HDL containing apoA-I and apoA-II (LpA-I:A-II) are distinctly different both structurally and metabolically [22]; LpA-I:A-II is more pro-atherogenic [23]. LpA-I is more cardioprotective in patients with coronary artery disease. Bioinformatic analysis revealed that Lp-A-I is more metabolically active and facilitates the increase of HDL particle size and number [21]. LpA-I:A-II is metabolically less active and usually has a smaller particle size than LpA-I [21]. Moreover, because apoA-II is more hydrophobic than apoA-I and, upon an increase in dimeric apoA-II concentration, it displaces apoA-I from the surface of HDL [24], it can result in a change of binding affinity with SARS-CoV-2 and SR-BI. 

The limitations of this study are different sources of HDL, different cell lines, different viral strains, and different assay systems were used in the two studies. It cannot be excluded that these differences may have affected the results of the studies. Furthermore, the precise role of apoA-II upon binding with SARS-CoV-2 and SR-BI has yet to be elucidated.

In conclusion, the elevated level of apoA-II might interfere with a putative interaction between HDL and SR-BI. More study on a reconstituted HDL containing either apoA-I or apoA-II alone will be needed to elucidate the precise binding mechanism of HDL and SARS-CoV-2 via the SR-BI. The composition and properties of apoA-I and apoA-II in HDL might be critical to SARS-CoV-2 infection via SR-BI and the antiviral activity against COVID-19.

## Data Availability

The data used to support the findings of this study are available from the corresponding author upon reasonable request.

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
