# Peer review of "Importance of Apolipoprotein A-I and A-II Composition in HDL and Its Potential for Studying COVID-19 and SARS-CoV-2"

_medicines, 2021, doi:10.3390/medicines8070038_

Round 1

Reviewer 1 Report

The author correlated between the data obtained by "Wei, C. et al. HDL scavenger receptor B type 1 facilitates SARS CoV 2 entry. Nat. Metab . 2 1391 1400 (2020)" and his data "Native High Density Lipoproteins (HDL) with Higher Paraoxonase Exerts a Potent Antiviral Effect against SARS CoV 2 (COVID 19), While Glycated HDL Lost the Antiviral Activity" and other clinical data. In the first article, the authors showed that HDL enhanced virus virus entry through SR-B1 through a mechanism not fully identified but might include interaction between HDL and the virus, although there was no clear data on stable interaction between HDL and the virus. In the second article, the author showed that native HDL has antiviral effect.  The argument that the author used to describe this contradiction "if any" is not convincing. In both studies different sources of HDL and different cell lines were used. It is well known that virus entry is different among the cell lines. Further, the applied methodology could also affect the outcome. The claim that apoA-II present in HDL preparation of the first study could modulate the interaction with or entry of SARS-cov2 lacks evidence and it is just hypothetical. The correlation between serum HDL-C level and the outcome of infections in clinical cases cannot be correlated with the in vitro studies giving the different setup of the assay. 

In all cases, this commentary is more hypothetical and lacks evidence to support the conclusion. I totally agree with the author that the different component of HDL (but also cell lines and methods used) might affect the outcomes of infection or entry. However, it would be convincing to show solid data to support this claim.

Author Response

I absolutely agree with reviewer’s opinion

Yes.  It is a possibility that presence of apoA-II in commercial HDL could affect viral entry because apoA-II can modulate the reaction of HDL via displacement of apoA-II.

Many reports suggested that presence of apoA-II modulates the binding and selective lipid uptake of HDL by SR-BI. These results might influence to different faciilitation of viral entry via SR-BI because apoA-2 showed different kinetic stabilization and binding affinity of HDL (Biophys. J. 2005;88:2907-2918) .   

There have been many reports to show that apoA-I and HDL exerted antiviral activity, while apoA-II did not show the antiviral activity.

   Agreed.  The sentences about lower serum HDL-C and outcome of infections were revised as below.

“Although the in vitro study cannot be directly compared with other clinical cases, on the other hand, these results are questioned by many clinical reports regarding the human lipid profile after the onset of COVID-19.”

  I removed hypothetical sentences.  This commentary is more focusing that apoA-II can interfere antiviral activity of apoA-I.

ApoA-II displays pro-inflammatory activity that may help maintain sensitive host responses to LPS by suppressing LBP-mediated inhibition.  ApoA-II does not inhibit LPS bioactivity in the presence of low level of LBP. Thompson, P.A.; Berbée, J.F.; Rensen, P.C.; Kitchens, R.L. Apolipoprotein A-II augments monocyte responses to LPS by suppressing the inhibitory activity of LPS-binding protein. Innate Immunity 2008, 14, 365-374

Thank you very much for your valuable comments to improve this commentary

Reviewer 2 Report

Dear Dr.  Jenny Lin,

In this study, entitled “ Importance of apolipoprotein A-I and A-II 2 composition in HDL and its quality to study COVID-3 19 and SARS-CoV-2”,

The manuscript was revised and in that version can be published to the journal.  

Author Response

Thank you very much for the comments and recommendation for acceptance.

Reviewer 3 Report

In this manuscript, the author emphasized the importance of apolipoprotein A-I and A-II composition in HDL and pointed out the unexpected composition of a commercial available HDL used in a previous study. This discovery can be quite significant for researchers using this product or reconsider previously published conclusions. But I found this manuscript is not very supported by scientific data. There is only one figure in the whole manuscript. And the figure is already published on a previous publication (Citation 9: Cho, K. H., Kim, J. R., Lee, I. C. & Kwon, H. J. Native High-Density Lipoproteins (HDL) with Higher 181 Paraoxonase Exerts a Potent Antiviral Effect against SARS-CoV-2 (COVID-19), While Glycated HDL Lost 182 the Antiviral Activity. Antioxidants (Basel). 10, 209 (2021)). This’s concerning. And The SDS-PAGE gel picture of the SIGMA HDL is supplied by the scientist. The authors guessed the components of the SIGMA product without doing any scientific confirmation, such as mass spectrum and NMR. Anyway, this manuscript lacks convincing scientific evidences. I don’t think this is a publishable manuscript.

Author Response

Thank you very much for the comments.  From this commentary, I also hope that many researchers can realize difference of composition depends on HDL in their study.  Commercial HDL may have different glycation extent and apolipoprotein compositions. Quality of HDL should be considered in prior to use and interpretation of results.

There are many functional differences between native HDL and glycated HDL in regarding anti-inflammatory activity, antiviral activity and receptor-mediated viral entry.  Antiviral activity of native HDL can be explained that it can involve direct viral inactivation, interference with viral entry, inhibition of virus-induced cell fusion (J. Cell Biochem.1991;45:224-237). However, glycated HDL from patients with type II diabetes lost the anti-inflammatory activity (Antiviral Res. 1999; 42: 211-218).

An employee of Research & Applied Solutions (Sunghye Kim) in Sigma-Aldrich   answered to confirm that it is apoA-II based on molecular weight determination as below email.  She is working in Sigma-Aldrich Korea. They approved the usage of gel photo in this commentary.

It is reasonable to conclude that the minor band around 10 kDa is ApoA-II, because apoA-II is dimeric protein with disulfide-linked 9 kDa of monomer. 

It has been well known that native HDL displays innate immunity, mediating several functions that defend against viral and bacterial infections.  apoA-I is antimicrobial protein in non-vaccinated rainbow trout model but not apoA-II (Villarroel F, Bastías A, Casado A, Amthauer R, Concha MI. Apolipoprotein A-I, an antimicrobial protein in Oncorhynchus mykiss: Evaluation of its expression in primary defence barriers and plasma levels in sick and healthy fish, Fish & Shellfish Immunology (2006), doi: 10.1016/j.fsi.2006.10.008

Round 2

Reviewer 1 Report

Thank you for submitting the revised version which has been improved significantly.

Reviewer 3 Report

I appreciate the authors revised the manuscript. I don't have more comments.